# Overlapping Community Detection Based on Membership Degree Propagation

**DOI:** 10.3390/e23010015

**Published:** 2020-12-24

**Authors:** Rui Gao, Shoufeng Li, Xiaohu Shi, Yanchun Liang, Dong Xu

**Affiliations:** 1Key Laboratory of Symbol Computation and Knowledge Engineering of the Ministry of Education, College of Computer Science and Technology, Jilin University, 2699 Qianjin Street, Changchun 130012, China; gaor17@mails.jlu.edu.cn (R.G.); lisf18@mails.jlu.edu.cn (S.L.); ycliang@jlu.edu.cn (Y.L.); 2Zhuhai Laboratory of Key Laboratory of Symbolic Computation and Knowledge Engineering of Ministry of Education, Zhuhai College of Jilin University, Zhuhai 519041, China; 3Department of Electrical Engineering and Computer Science, Informatics Institute, and Christopher S. Bond Life Sciences Center, University of Missouri, Columbia, MO 65211, USA

**Keywords:** complex network, social network, overlapping community detection, label propagation, membership degree, clustering

## Abstract

A community in a complex network refers to a group of nodes that are densely connected internally but with only sparse connections to the outside. Overlapping community structures are ubiquitous in real-world networks, where each node belongs to at least one community. Therefore, overlapping community detection is an important topic in complex network research. This paper proposes an overlapping community detection algorithm based on membership degree propagation that is driven by both global and local information of the node community. In the method, we introduce a concept of membership degree, which not only stores the label information, but also the degrees of the node belonging to the labels. Then the conventional label propagation process could be extended to membership degree propagation, with the results mapped directly to the overlapping community division. Therefore, it obtains the partition result and overlapping node identification simultaneously and greatly reduces the computational time. The proposed algorithm was applied to a synthetic Lancichinetti–Fortunato–Radicchi (LFR) dataset and nine real-world datasets and compared with other up-to-date algorithms. The experimental results show that our proposed algorithm is effective and outperforms the comparison methods on most datasets. Our proposed method significantly improved the accuracy and speed of the overlapping node prediction. It can also substantially alleviate the computational complexity of community structure detection in general.

## 1. Introduction

Complex networks are natural manifestations of many real-world problems, such as social networks, computer networks, and protein interaction networks. Although a long history of complex network development exists, a sharp increase in related problems and data has ushered in a more extensive development of these networks in recent years. Small world [1] and scale-free [2] properties, as well as high aggregation [3] are the most obvious characteristics of complex networks. The aggregation feature is often measured according to the community structure of a network [4,5]. In complex networks, a community refers to a group of nodes that are densely connected internally but sparsely connected to the outside. Generally, the nodes belonging to the same communities have similar functions or properties and vice versa. For example, the nodes in a same community of social network often indicate that they might have a same family, a same career, or a same hobby [6], while those of a protein–protein interaction network are probably proteins with similar functions [7]. Through studying the community structure of a complex network, we can better understand the network nature as a whole and different functions as local communities as well.

In a complex network, there are interactions between different communities with an important form wherein different communities share the same nodes. We call these nodes overlapping nodes, and the communities are referred to as overlapping communities. Overlapping nodes and overlapping communities exist widely in complex networks in the real world. For example, one individual may be in multiple communities (e.g., families) of a social network. In the biomolecular network, different communities can represent different biological functions, and a gene or protein can participate in a variety of biological functions. In the academic circle, a scholar often works in multiple fields. Overlapping nodes often play an important role in complex networks. Because overlapping nodes belong to and connect multiple overlapping communities and play a pivotal role in information flow, identification of overlapping nodes is an important research topic in complex network analyses. For example, Mengoni et al. have studied student population community elicitation, and found that the co-occurrence of people’s activities is an emerging epiphenomenon of hidden, implicit exchanges of information in side-channel communications [8]. Many researchers have investigated the importance of overlapping nodes in epidemic spreading, and then developed immunization strategy accordingly [9,10,11]. 

In 2002, Girvan and Newman proposed the well-known Girvan–Newman (GN) algorithm [4], which defines the concept of edge betweenness and holds that the edge betweenness within a community should be smaller than the edge betweenness between communities. Since then, many community detection algorithms have been proposed, details can be found in Liu et al.’s review of community mining in complex networks [12]. However, traditional non-overlapping community detection algorithms cannot be directly applied to overlapping community detection; hence various overlapping community detection algorithms have been developed, which could be classified into seven categories: (1) clique percolation, (2) link partitioning, (3) local expansion and optimization, (4) fuzzy detection, (5) matrix (tensor)-based model, (6) statistical inference (7) label propagation. For comprehensive overview, one can refer to [13,14].

*Clique percolation method* (CPM) holds that the inner edges of a community are closely connected with each other and have high edge density; thus, it is easier to form cliques (complete subgraphs) within communities [15]. In CPM, communities consist of those cliques being strongly connected with each other and overlapping nodes are recognized if they belong to multiple cliques assigned to different communities [16]. Cui et al. have extracted fully connected sub-graphs using maximal sub-graph method [17]. *Link partitioning* [18,19] is based on edges to find the community structure. If a link is put in more than one cluster, then the nodes this link connects to are labeled as overlapping nodes [20]. Arasteh M and Alizadeh S proposed a fast divisive community detection algorithm based on edge degree betweenness centrality [21]. A classical *local expansion and optimization* model is local fitness model (LFM) [22], which starts from a random seed node and extends the community step by step until the fitness function is locally maximized. Subsequently, LFM randomly selects a node that is not in the generated communities as a new seed node and repeats the expansion of the community until all nodes belong to one or more communities. Then, those nodes belonging to multiple communities are considered as overlapping nodes. Following the idea of LFM, the greedy clique expression (GCE) [23] selects the maximal clique as the seed. Guo K et al. proposed a local community detection algorithm based on internal force between nodes to extend the seed set [24]. Zhang J et al. proposed a series of seed-extension-based, overlapping, community detection algorithms to reveal the role of node similarity and community merging in community detection [25]. Eustace et al. have utilized neighborhood ratio matrix to detect local communities [26]. Other local expansion and optimization models include OSLOM [27], Infomap [28], Game [29], and so on. *Fuzzy detection* [30] calculates the connection strength between each pair of nodes and between communities; it also assigns a membership vector to each node. The dimension of membership vectors must be determined, as they can be used as algorithm parameters or calculated from data. *Matrix(Tensor)-based model* represents the network structures by matrix or tensor, and yields a more robust community identification in the presence of mixing and overlapping communities by matrix factorization [31] or tensor decomposition [32]. *Statistical inference* can effectively tackle the problem of community detection and has many useful methods like: MMSB, AGM and BIGCLAM et al. MMSB, which combines global parameters that instantiate dense patches of connectivity (blockmodel) with local parameters that instantiate node-specific variability in the connections (mixed membership), can be used in overlapping community detection [33]. AGM is a community-affiliation graph model that builds on bipartite node-community affiliation networks [34]. BIGCLAM (cluster affiliation model for big networks) is an overlapping community detection method which scales to large networks of millions of nodes and edges [35].

The final widely used type of overlapping community detection algorithm is based on *label propagation.* Its main idea is to assign a label to represent its class for each node, and to propagate the label messages according to network structure and the label distribution until it is converged. After the label propagation, the nodes in the same community are assigned a same label. The classical label propagation algorithm (LPA) [36] was developed for non-overlapping community detection. Because of LPA, several groups have extended the method into overlapping community detection. For example, community overlap propagation algorithm (COPRA) [37] allows assignment of multiple labels for each node, associated with belonging coefficients to indicate the strengths of memberships for different classes. In contrast to COPRA, Xie et al. proposed another label propagation algorithm, which spreads labels among nodes during iterations and saves previous label information for each node [38]. Le B D et al. proposed an improved network community detection method using meta-heuristic based label propagation [39]. Because the label propagation process is just like that of speaker-listener communication, the algorithm is referred to as the speaker–listener label propagation algorithm (SLPA). Based on SLPA, Gaiteri et al. proposed the SpeakEasy algorithm [40], which introduced the label global distribution information into label propagation, and effectively superseded the local neighborhood label information. 

SpeakEasy as mentioned above is suitable for several different kinds of networks, and has good adaptabilities compared to previous algorithms. However, the threshold to identify overlapping nodes is difficult to determine, which might result in poor recognition of overlapping nodes. Furthermore, SpeakEasy requires generating community partitions many times to obtain a robust result, which is computationally time consuming.

To address the above weaknesses, this paper proposes a new overlapping community detection algorithm. In our method, it is the membership degree being propagated rather than the label information in existing label propagation algorithms, and therefore is called membership degree propagation algorithm (MDPA). The membership degree represents the probability that a node belongs to a potential community, which replaces SpeakEasy’s sampling of community partitions. Our method is different from COPRA since the latter propagates label information and gives a belonging coefficient for each label. MDPA has been applied to a Lancichinetti–Fortunato–Radicchi (LFR) artificial dataset and to nine commonly used real datasets. Numerical results show that MDPA greatly improves the recognition of overlapping nodes in accuracy and speed. The main contributions of the paper are (1) the introduction of the concept of membership degree, which not only stores the label information, but also the membership degree of the node belonging to the label; and (2) the significant reduction computing cost for that MDPA does not need replaying the algorithm to achieve the overlapped community partition.

## 2. Introduction of Label Propagation Methods

The methods based on label propagation have been greatly developed and widely used. Generally speaking, the methods based on label propagation consist of four parts, namely that initialization, label propagation, community partition and overlapping node identification, the general framework is shown as Figure 1. In initialization, the label propagation methods usually need to assign a buffer to each node to store the label information. The size of the buffer can be fixed or not fixed, but it must have the maximum size. In label propagation, the methods usually iterate many times to make labels propagate in the whole network. Finally, it comes to convergence. In order to judge whether it is convergence or not, it is necessary to judge the difference between the current network and the network after the latest iteration. It’s simpler to iterate enough times which will consume a lot of computing time. In community partition, at least one label should be assigned to each node. Some methods will divide communities according to the label information of its own buffer, while others will be based on the label information of neighbor nodes’ buffer. In overlapping node identification, the basic idea is that a node will be determined as an overlapping node if it belongs to two or more communities. Different methods have different ways of discrimination.

As mentioned in Section 1, MDPA is a natural extension of the existing label propagation methods, such as LPA, COPRA, SLPA, and most notably, SpeakEasy. Therefore, SpeakEasy is taken as an example to describe the details of Label propagation methods in the following of this section. Figure 2 shows the flowchart of SpeakEasy with four steps, which are described as follows: *Initialization.* To initialize the node buffers of the whole network, the ID numbers of all nodes are set as potential community labels initially. Each node’s ID number is pushed into its own buffer at the beginning. Then, the neighbor ID numbers will be randomly selected to fill the buffer of each node, until it is fully filled.*Label propagation.* To propagate the labels iteratively, each node should update its buffer by pushing in the most “significant” label in its neighbor buffers and pushing out the first one at the beginning. The “significance” of a label is determined by the difference of its distribution in the local neighborhood buffer set and the distribution in the global buffer set. In other words, the more it is in the local buffer set, and the less it is in the global buffer set, the more “significant” the label is. This process is performed iteratively, until it is converged.*Community partition of single round.* After the label propagation process is converged, each node should be assigned the community ID number by most labels in its neighbor node buffers. Then, we will get a community partition *P*.*Common community partition.* Repeat step 1 to step 3 for N times to obtain *N* candidate partitions {*P*_1_, *P*_2_, ∙∙∙, *P_N_*}. The partition that is most similar to others will be selected as the final nonoverlapping community partition, denoted as *P** = {*C*_1_*, *C*_2_*, ∙∙∙, *C_K_**}, where *C_i_** is the ith cluster in partition *P**. This is a more robust result than that of only one iteration for a nonoverlapping problem. If the problem is for nonoverlapping community detection, the algorithm can stop here; otherwise, it will continue to step 5.*Overlapping node identification.* Denote *a_ij_* as the number of times nodes *v_i_* and *v_j_* cluster together in the obtained *N* partitions, and let *a_ii_* = 0; then, the co-occurrence matrix A can be constructed. Assuming node *v_i_* is not a member of the *j*th cluster *C_j_** in partition *P**, define the weight of *vi* to *C_j_** by:
(1)$$w_{v_{i}}^{C_{j}^{*}}=\frac{\sum_{u\in C_{j}^{*}}a_{u,v_{j}}}{\left|C_{j}^{*}\right|\cdot N}$$
where *N* is the number of partitions, and |·| means the size of a cluster. If the weight is big enough, node *v_i_* is considered an overlapping node of cluster *C_j_**. In [16], the threshold is set as 1/*K*_max_, where *K*_max_ is the largest number of communities of *N* partitions. However, in our experiments, sometimes this threshold resulted in too many overlapping nodes, making it difficult to adjust the threshold.

## 3. Membership Degree Propagation Algorithm

In the SpeakEasy algorithm, the label propagation process needs to be repeated *N* times, which greatly reduces the efficiency when the network size is large. A more important problem is that the threshold to identify overlapping node is difficult to choose, which often leads to an improper proportion of the overlapping nodes. In view of the above problems, a new algorithm is proposed in this paper. The main idea of the algorithm is to define a membership degree vector for each node representing how likely it belongs to the potential clusters, and then propagate the membership degree vector instead of the label in the existing label propagation methods. Hence, the algorithm is called membership degree propagation algorithm (MDPA), which is roughly divided into three steps: (1) initialization, (2) membership degree propagation, and (3) community partition. The flowchart of MDPA is shown in Figure 3. The framework of MDPA does not have the outer loop in Figure 2, and the overlapping node identification is merged with the community partition. The details of the algorithm are illustrated as follows.

### 3.1. Initialization Process

For simplicity, we will only discuss the undirected, unweighted graph below. It is easy to extend to a directed or weighted graph. Let ***V*** = {*v*_1_, *v*_2_, ∙∙∙, *v_n_*} be the set of vertices (or nodes); ***E*** is the set of edges, each representing a pair of nodes (*x*, *y*) ϵ ***V*^2^**, meaning that there is an edge between nodes *x* and *y*. Now we can find the overlapping community partition on graph ***G*** = {***V***, ***E***}.

Much like SpeakEasy and other existing label propagation algorithms, we construct a buffer for each node. However, the difference is that the buffer not only stores the label information, but also the membership degree of the node belonging to the label. So, an element of the buffer is a binary group. Here, we define membership as the possibility that the current node belongs to a potential community. Thus, we denote the buffer of the *i*th node *v_i_* as: (2)$$b_{i}=\left\{\left(l_{1}^{\left(i\right)},m_{1}^{\left(i\right)}\right),\left(l_{2}^{\left(i\right)},m_{2}^{\left(i\right)}\right),\cdots ,\left(l_{B^{\left(i\right)}}^{\left(i\right)},m_{B^{\left(i\right)}}^{\left(i\right)}\right)\right\},\;\left(l_{2}^{\left(i\right)}\in \{1,2,\cdots ,n\},\;B^{\left(i\right)}\leq B\right)$$
where $l_{j}^{\left(i\right)}$ represents a potential cluster of node *v_i_*, and $m_{j}^{\left(i\right)}$, and the corresponding membership degree of node *v_i_* belongs to cluster *l_j_*, which should satisfy
(3)$$\sum_{j}m_{j}^{\left(i\right)}=1$$

The constant value *B* in Equation (2) represents the maximum number of potential clusters for each node, which is set to 3 times the average node degree in our experiments. 

Like SpeakEasy, the ID numbers of all nodes are set as potential community labels initially. For each node, its own ID number is pushed into the label part of buffer at the beginning, and the membership degree is set as 1/*B*. Then, its neighbor ID numbers will be randomly selected for *B*-1 times, with the membership degree adding 1/*B* correspondingly. It should be noted that if an ID number is selected more than one time, its membership degree will be more than 1/*B*, and its buffer length will then less than *B*. Figure 4a shows an initialization example of a simple network with seven nodes, where the parameter *B* is set as 5.

### 3.2. Membership Degree Propagation

The main idea of the membership degree propagation process is to increase the membership degree of the clusters with higher local distribution and lower global distribution, and vice versa. For a cluster *c*, the global distribution is its appearing frequency in the entire network buffers, which is calculated by:(4)$$g_{c}=\frac{\sum_{j}m_{c}^{\left(j\right)}}{n\cdot B}$$
The local distribution of cluster *c* for node *v_i_* is the occurrence frequency in its neighbor node buffers, which is calculated by:(5)$$f_{c}^{\left(i\right)}=\frac{\sum_{j\in neighbor(i)}m_{c}^{\left(j\right)}}{\left|neighbor(i)\right|\cdot B}$$
Taking Figure 4a as an example, the global distribution is listed in Table 1, and the local distribution of node *d* is listed in Table 2.

With these tables in place for each node, we now calculate the difference between local distribution and global distribution for each cluster ID in its neighbor buffers. For the calculation to make sense, it should be normalized into a same scale; in our experiments, the scale is set as [0, 5]. Therefore, the normalized difference of cluster *c* for node *v_i_* is computed by:(6)$$d_{c}^{\left(i\right)}=\alpha \cdot \frac{\left(f_{c}^{\left(i\right)}-g_{c}\right)-\underset{j}{\min}\left(f_{j}^{\left(i\right)}-g_{j}\right)}{\underset{j}{\max}\left(f_{j}^{\left(i\right)}-g_{j}\right)-\underset{j}{\min}\left(f_{j}^{\left(i\right)}-g_{j}\right)}$$
where *α* is the scale parameter, set as 5 in our experiments. Then, cluster *c* will be selected to update the buffer of node *v_i_* according to the probability:(7)$$p_{c}^{\left(i\right)}=\frac{e^{d_{c}^{\left(i\right)}}}{\sum_{c\in neighbor(i)}e^{d_{c}^{\left(i\right)}}}$$
Still taking node *d* in Figure 4a as an example, the original difference, the normalized difference, and the corresponding probabilities of each label are listed in Table 3. 

Next, we randomly select a cluster from the buffers of node *v_i_* according to the probability of Equation (7). If the selected cluster has already existed in the buffer *b_i_*, the corresponding membership degree adds 1/*B*, and the other clusters’ membership degrees are adjusted to let the sum remain equal to 1. Otherwise, the selected cluster should be added into the buffer *b_i_* with the membership degree of 1/*B*. If the buffer length is larger than *B*, the cluster with the smallest membership degree is removed. However, if there are many clusters with the smallest membership degree, randomly delete one. The membership degrees are then adjusted to let the sum remain equal to 1. In each iteration, all nodes in the network need to update its buffer as described above. When the processing is converged or the number of iterations reaches its limit, the loop stops. The membership degree propagation pseudo-code is shown as Algorithm 1.
**Algorithm 1**:* Membership Degree Propagation* (*B, NUM, G’*)
**Input**: Buffer Size: *B*, The number of iterations: *NUM*, Initialized graph: *G’*
**Output**: The convergent graph after *Membership Degree Propagation*: *G’*1*i* = 0; 2**for***i* < *NUM*
**do**3 **for**
*v* ∈ *G’*
**do**4  *g* ← Calculate the probability distribution of all labels in the current network;5  *f*^(*v*)^ ← Calculate the probability distribution of all labels in the local subgraph of the current node *v*;6  *L* ← Get the maximum difference label *L* according to *d*^(*v*)^, where *d*^(*v*)^ = *f*^(*v*)^ − *g*;7  *G’’* Increase the membership degree corresponding to label *L* and update the buffer of *v*;8 **end**9 *i* = *i* + 1;10**end**

For node *d* in Figure 4a, suppose the randomly selected label is *d* according to the probabilities computed as Table 3. The result after updating the buffer of node *d* is shown in Figure 4b, and the final result is shown in Figure 4c.

### 3.3. Community Partition

At the end of membership degree propagation process, we obtain the buffer sets for all the nodes in the network. A buffer contains no more than *B* binary groups, each of which represents a potential cluster and its membership degree correspondingly. Then, the community partition can be divided into two simple steps:Assign the max membership cluster in the buffer as the first community for each node
(8)$$v_{i}\leftarrow \underset{c}{\arg}m_{c}^{\left(i\right)}$$
where *v_i_* is the *i*th node, and $m_{c}^{\left(i\right)}$ is the membership of *v_i_*, which belongs to cluster *c*.Identify the secondary communities for each node. For node *v_i_*, we make the traversing of its buffer as follows:(9)$$v_{i}\leftarrow c\;if\;m_{c}^{\left(i\right)}>r$$
where *r* is a membership threshold, which is simply set as *r* = 1/*N*l, where *N*l is the account number of the first communities assigned to nodes in the above step. Therefore, N1 can be considered as the initial number of communities assigned to all nodes. Then, according to Equation (9), if the membership $m_{c}^{\left(i\right)}$ is larger than the probability randomly assigns a node to any a community, node *i* is considered as a member of cluster *c*. 

This process is simple, and it is easy to recognize overlapping nodes just by checking to see if more than one cluster have been assigned to them. Therefore, MDPA does not need to repeat the outer loop (initialization and propagation) multiple times, which reduces computational cost greatly when compared to SpeakEasy. The pseudo-code of the community partition is shown as Algorithm 2. The final community partition result is shown in Figure 4d.
**Algorithm 2**: CommunityPartition (r, B, G″)
**Input**: Threshold: *r*, Buffer Size: *B*, The convergent graph after *Membership Degree Propagation*: *G″*
**Output**: The result of community detection: *C*1**for***v* ∈ *G″*
**do**2 *flag* = 0;3 **for** (*l*^(*v*)^, *m*^(*v*)^) ∈ *b_v_*
**do**4  *// l*^(*v*)^ represents a potential cluster of node *v*;5  *//m*^(*v*)^ is the membership degree of node *v* belonging to cluster *l*^(*v*)^;6  **if**
*m*^(*v*)^ > r **then**7   *l*^(*v*)^ ← *l*^(*v*)^ ∪ {*v*};8   *C* ← *C* ∪ {*l*^(*v*)^};9    *flag* = 110  **end**11 **end**12 *//m*^(*v*)^ <= *r* is true for all *m*^(*v*)^;13 *//*The node *v* is an overlapping node and belongs to all the clusters in buffer *b_v_*14 **if**
*flag* = 0 **then**15  **for** (*l*^(*v*)^, *m*^(*v*)^) ∈ *b_v_*
**do**16*l*^(*v*)^ ← *l*^(*v*)^ ∪ {*v*};17    *C* ← *C* ∪ {*l*^(*v*)^};18**end**19**end**20**end**

### 3.4. Complexity Analysis

In the initialization phase, MDPA needs to traverse the whole network and fill the membership degree of each traversed node’s buffer. Each node needs to be filled B times, so it needs *nB* operations:
*N_l_* = *nB*(10)

In the propagation phase, it is assumed that *N* iterations are needed and adjusting each buffer needs fixed *A* operations. For the current node *v* traversed, we need to calculate the global probability distribution at first, and the number of required operations is *nB*. Then we calculate the local probability distribution of node *v*. It needs at most $N_{nei}^{v}\times B$ operations, where $N_{nei}^{v}$ is the number of neighbor nodes of node *v*. Finally, the buffer is adjusted through *A* operations. Therefore, the maximum number of operations MDPA needs to perform in this process:(11)$$N_{p}=N\times \sum_{v}\left(nB+N_{nei}^{v}B+A\right)$$

We can see that *n* is much larger than $N_{nei}^{v}$, and the change of global probability is only due to the update of the buffer of one node. After adjusting the buffer of one node each time, the new global probability distribution can be obtained by only a few operations. Here, the number of operations to adjust the global probability can be regarded as a fixed number of operations. Based on this idea, after optimizing the algorithm, the maximum number of operations needed for MDPA in the propagation phase is as follows:(12)$$N_{p}=nB+N\times \sum_{v}\left(N_{nei}^{v}B+A+A_{g}\right)=nB+2mNB+nNA+nNA_{g}$$
where *A_g_* is the fixed number of operations required to adjust the global probability distribution. In other words, at the beginning of the propagation process, we calculate the global probability distribution, and then adjust the global probability distribution after completing the buffer adjustment of the current node.

In the overlapping node identification stage, MDPA only needs to traverse each node and compare the membership degree of each node buffer with the threshold *r*, so MDPA needs *nB* operations at most in this stage.
*N_o_* = *nB*(13)
So, the maximum number of operations required by MDPA is as follows:(14)$$N_{total}=N_{l}+N_{p}+N_{o}=3nB+2mNB+nNA+nNA_{g}$$

For space complexity, since the buffer allocated to each node in the network contains up to *B* potential communities, MDPA takes up *nB* storage units at most.

## 4. Experiments and Results

### 4.1. Experiment Setup

In this paper, two types of experiments are designed to test the proposed MDPA method, one for the LFR benchmark dataset [41], and the second for real dataset. 

The LFR benchmark provides many parameters to control the structure of the generated network. Using different parameter values, we have generated 180 networks for experiments. To test the performance of the proposed MDPA, four state-of-the-art methods are executed for comparison, namely, SLPA, Olsom, Copra, and SpeakEasy, respectively. The LFR benchmark has many advances, for example, it has clear ground truth for evaluation, and its parameters can be set flexibly. However, it still has some limitations: it is more suitable for generating medium-sized networks and therefore cannot meet the needs of large networks for experiments; it is difficult to analyze theoretically because of the complexity of the algorithm; and it is far from generating very large realistic artificial networks [42]. Therefore, we also applied our proposed method to real dataset for validation.

For the real data set, we conducted MDPA on the nine datasets (listed in Table 5). For the comparison, 8 state-of-the-art algorithms are also executed on those datasets, which are SpeakEasy, Perception, SLPA, Ego, Angel, Demon, Kclique, and LFM, respectively.

All of the experiments were executed on a PC with a 2.40 GHz Intel(R) Xeon(R) CPU, 16GB memory, and the Windows 7 Ultimate 64-bit operating system. We used Java to implement the code and the programming environment was Eclipse.

### 4.2. Evaluation Metrics

There are many methods to evaluate a partition result, however, each of them has its applicability and limitations. In the LFR benchmark dataset, the ground-truth are known, therefore the similarity of partition result with ground-truth could be used for evaluation. NMI (normalized mutual information) [22] is commonly used metrics to measure the similarity of two partitions. For a network with *n* nodes, assuming that there are two partitions: *P* = {*p*_1_, *p*_2_, …, *p_I_*}, and *G* = {*g*_1_, *g*_2_, …, *g_J_*}, where *I* and *J* are the community numbers of *P* and *G*, the NMI between *P* and *G* is defined by Equation (15): (15)$$NMI(P,G)=1-\frac{1}{2}\left[H{\left(P|G\right)}_{norm}+H{\left(G|P\right)}_{norm}\right]$$
where *H*(*P*|*G*)*_norm_* (or *H*(*G*|*P*)*_norm_*) is the normalized conditional entropy of *P* (or *G*) with respect to *G* (or *P*). For more details of NMI formula for overlapping communities, please refer to Appendix B in reference [22].

NMI works well to validate the functionality of proposed methods in ad-hoc networks but are not directly interpretable in a comparative evaluation of clustering quality. The metric of S_G_ evaluates the similarity of two partitons is to inquire into the “closeness” of the two corresponding community size distributions [43]. Assume there are r different sizes of communities in partition P, and arrange sizes ascendly as {*x*_1_*^P^*, *x*_2_*^P^*, …, *x_r_^P^*}, and those of partition *G* are {*x*_1_*^G^*, *x*_2_*^G^*, …, *x_s_^G^*}. Then *S_G_* could be defined by:(16)$$S_{G}(P,G)=\frac{1}{2}\sum_{i=1}^{r}\sum_{j=1}^{s}\min\left\{\frac{n^{P}\left(x_{i}^{P}\right)}{N^{P}},\;\frac{n^{G}\left(x_{j}^{G}\right)}{N^{G}}\right\}\delta \left(x_{i}^{P},x_{j}^{G}\right)$$
where *n^P^*(*x_i_^P^*) means the community number with size of *x_i_^P^* in partition *P*, and similarly for *n^G^*(*x_j_^G^*), *NP* and *NG* are the total number of communties in partions *P* and *G*, *δ*(*x_i_^P^*, *x_j_^G^*) = 1 if *x_i_^P^* = *x_j_^G^* and 0 otherwise. *S_G_* evaluates high level of clustering similarity between two partitions, but can not detect similarity on iner-community level. It is better to combine it with NMI to evaluate the similarity of two partitions. 

Since overlapping nodes generally play key roles in overlapping community networks, it is important to predict overlapping nodes correctly in the overlapping community detection. In this sense, the overlapping node detection could be considered as a binary classification problem. Recall, Precision and F1 measures are commonly used metrics in classification problems, as defined in Equations (17)–(19):
Recall = *TP*/(*TP* + *FN*)
(17)

Precision = *TP*/(*TP* + *FP*)
(18)

F1 = 2 × (Precision × Recall)/(Precision + Recall)
(19)
where *TP* is the number of positive samples predicted as positive samples; *FN* is the number of positive samples predicted as negative samples, and *FP* is the number of negative samples predicted as positive samples. The F1 measure is a more comprehensive metric, which combines Recall and Precision together.

In real data sets, the ground-truth is often considered as unknown. Under this condition, the goodness of a partition could be verified by characterizing how community-like the connectivity structure of partition is. The idea is to that a “good” partition has dense connections with communities and sparse connections between different communities. There are many ways to determine whether a partition meets the above criteria, for example, based on internal connectivity, based on external connectivity, combining internal and external connectivity, and based on network model [44]. They all have their own advantages, but also have different limitations. Conductance is a metric combining internal and external connectivity, defined by:(20)$$\mathrm{Conductance}(P)=\frac{c_{P}}{2m_{P}+c_{P}}$$
where *m_P_* is the total number of edges within single communities in partition *P*, and *c_P_* is the total number of edges between different communities in partition *P*, respectively. Thus, metric Conductance measures the fraction of total edge volume that points outside the cluster [45].

Modularity *Q* [46,47] is a well-known metric based on network model, defined by:(21)$$Q=\frac{1}{2m}\sum_{i,j}\left(w_{ij}-\frac{k_{i}k_{j}}{2m}\right)\delta \left(l_{i},l_{j}\right)$$
(22)$$\delta \left(x,y\right)=\{\begin{array}{cc} 1 & x=y\\ 0 & otherwise \end{array}$$
where *m* is the number of edges of the whole network (then 2*m* is the sum of degrees in the undirected graph), *w_ij_* the weight of edge between node *i* and *j*, being 1 or 0 for undirected networks. *k_i_* is the degree of node *i*, *l_i_* is the community ID that node *i* belongs to. Modularity Q measures the difference between the number of edges between nodes within single communities and its expected number in a random graph with identical degree sequence. However, it doesn’t consider the impact of overlapping nodes. Modularity EQ [48] is a variant for overlapping community detection, defined by:(23)$$EQ=\frac{1}{2m}\sum_{i,j}\frac{1}{O_{i}O_{j}}\left(w_{ij}-\frac{k_{i}k_{j}}{2m}\right)\delta \left(l_{i},l_{j}\right)$$
where *O_i_* is the number of communities containing node *i*. Modularity EQ is more suitable for overlapping community detection, and we use it as the evaluation metric together with Conductance in real data sets. 

### 4.3. Experiments on LFR Benchmark Dataset

To verify the performance of the proposed MDPA, it is executed on 180 LFR generated networks. For comparison, four state-of-the-art methods, Oslom, SLPA, Copra and SpeakEasy are also executed on the same data sets. The 180 generated networks are combined by the following parameter sets: *n* = {1000, 2000, 3000}, *d* = {10, 20}, *O_m_* = {1, 2, 4, 6, 8}, *μ* = {0.05, 0.1, 0.2, 0.3, 0.4, 0.6}, and *O_n_* = {30, 90, 150}, which are listed in Table 4.

Figure 5 shows the NMI comparison of our simulation results of MDPA and those of Oslom, SLPA, Copra and SpeakEasy. Figure 5a shows the average NMI values of different *μ*, *O_n_* and *O_m_* for specified *n* and *d* combinations, and Figure 5b shows the average NMI values of different *n*, *d*, *O_n_* and *O_m_* for specified *μ* values. From the figures we could find that OLSOM and MDPA are significantly superior to the other three algorithms, and OLSOM is slightly better than MDPA. With the parameter *μ* increases, the networks are changing to highly overlapping, and the performance of all the methods decreased, especially when *μ* is larger than 0.5, the performance dropped sharply. The detail comparison results of NMI on 180 LFR datasets could be inferred to Figure A1 in Appendix A.

Figure 6 shows the *S_G_* comparison results. Figure 6a shows the average *S_G_* values of different *μ*, *O_n_* and *O_m_* for specified n and d combinations, and Figure 6b shows the average *S_G_* values of different *n*, *d*, *O_n_* and *O_m_* for specified *μ* values. The results are similar with those of NMI, OLSOM and MDPA are much better than the other three algorithms, and OLSOM is slightly better than MDPA. The detail comparison results of *S_G_* on 180 LFR datasets could be inferred to Figure A2 in Appendix A.

To further investigate the performance of detecting the overlapping nodes, MDPA, Oslom, SLPA, Copra and SpeakEasy algorithms are also compared on F1 measure. Figure 7 shows the comparison results of the five algorithms on the F1 measure. Those missing points in the graph mean that the algorithm could not obtain a feasible F1 value, namely that no overlapping nodes were correctly found. It is easy to see that SLPA, Olsom and Copra are very unstable, they have 15, 12 and 70 missing points out of 144 (when *O_m_* = 1 means there is no overlapping nodes, therefore, only *O_m_* = 2, 4, 6, 8 are evaluated for F1 measure), respectively, while SpeakEasy and MDPA have no missing points at all. The detail comparison results could be inferred to Figure A3 in Appendix A. Compared with Speakeasy, our proposed MDPA algorithm is significantly better on almost all the networks with different setup parameters, where the only exception is the point of *O_m_* = 2 on the case of *n* = 1000, and *μ* = 0.4. Taking *n* = 3000, *d* = 10, and *μ* = 0.4 as an example, the F1 measures of MDPA are 0.21, 0.32, 0.37, and 0.35 respectively for *O_m_* = 2, 4, 6, and 8, while those of SpeakEasy are only 0.06, 0.07, 0.07, and 0.07. The main reason is that SpeakEasy tends to predict too many overlapping nodes thereby obtaining high recall values but too low of precision values, resulting in really low F1 measures, while MDPA could get a better balance of precision and recall values by predicting the proper number of overlapping nodes. 

The main factors affecting the running time of the algorithms are the network size (node number *n*) and average degree (*d*). Figure 8 shows the average executing times on each pair of *n* and *d* of the five compared algorithms. It clearly shows that MDPA runs slightly faster than Olsom, and much faster than SpeakEasy, especially when the networks were complex. The main reason is that SpeakEasy repeats the propagating process many times while MDPA executes it only once. Among the five algorithms, Copra and SLPA are the most fast two ones, the main reason is that they don’t execute multi-round iterations of the label propagation process, while it also results in high instability. 

### 4.4. Experiments on Real Benchmark Datasets

In order to further examine our proposed MDPA method, we apply it to nine commonly used real benchmark datasets. For comparison, eight state-of-the-art methods, namely, SpeakEasy [40], Perception [49], SLPA [39], Ego [32], Angel [50], Demon [51], Kclique [15], and Lfm [22] are also executed on the same datasets. Detailed information on the datasets is listed in Table 5.

Figure 9 shows the comparison results of MDPA with eight state-of-the-art methods on Conductance and EQ metrics. It could be found that SpeakEasy, Ego, kclique and Lfm can not obtain the reasonable results always on those “big” real datasets, namely, loc_brightkete, Dblp and Amazon datasets. Of the other 5 algorithms, MDPA, angle and demon obtains twice of rank 1 positions on Conductance metric (Figure 9a), respectively. While on modular EQ, MDPA obtains three times of rand 1 position, one less than SLPA, both of them are significantly superior to the other algorithms (Figure 9b). Table 6 shows the average Conductance and EQ values of different real datasets for all the nine algorithms. For Conductance, angle and MDPA get 0.447 and 0.444 average values and occupy the top 2 positions, which are significantly better than other algorithms. While for EQ, SLPA and MDPA are top 2 methods with values of 0.552 and 0.545, more than 30% higher than that of the third method (perception, 0.414). On the other hand, angle obtained the best performance on Conductance, but its average EQ value is only 0.319, far less than SLPA and MDPA. Similarly, though SLPA is the best one of average EQ, its average Conductance is only 0.231, about half of those of angle or MDPA.

### 4.5. Analysis

In this section, we will select each case from LFR and real datasets for further analysis on the obtained results of MDPA and SpeakEasy. For the LFR dataset, we take the case of *n* = 3000, *d* = 10, *μ* = 0.4 and *O_m_* = 6 as an example. Figure 10 shows the partition results of both SpeakEasy and MDPA, revealing that the community partition generated by MDPA is closer to the standard community partition than that by SpeakEasy. The local partition result of MDPA in Figure 10c is almost the same with its corresponding part in the ground truth of Figure 10b, while that of SpeakEasy in Figure 10d is quite different from the ground truth.

Overlapping node identification is most important for overlapping community detection. Table 7 shows the confusion matrix for the two algorithms. From the confusion matrix, it is easy to see that too many non-overlapping nodes are recognized as overlapping ones by SpeakEasy, while MDPA performs much better. Figure 11 shows the distribution of the weight defined by Equation (1) in SpeakEasy. SpeakEasy repeats the partition loop many times to find the consensus partition and the overlapping nodes as well. Due to randomness, the results are different each time, which results in a large amount of weights between 0 and 1, especially in the interval close to 0. According to the rule described in SpeakEasy [40], the weight threshold should be taken as 0.0057 (denoted as *r* in Figure 11). However, in a very wide neighborhood of *r*, the distribution of weights is evenly and not significantly different. That means it is difficult to find a better threshold, which determines whether a node belongs to another community. In fact, SpeakEasy recognizes too many nodes as overlapping in this case. It should be noted that we diligently sought a better way to determine the weight threshold, but it is difficult to find a rule that consistently works well for different instances. By propagating membership vectors instead of the node labels in SpeakEasy, MDPA can obtain the partition result and overlapping node identification simultaneously, which greatly reduces the computational time and avoids the difficulty of determining weight threshold in SpeakEasy.

For real benchmark datasets, the Football dataset was taken as the example for further analysis. There are 115 teams in the network represented by nodes and the teams are divided into 12 leagues, which can be considered ground truth for community partition. Figure 12 shows the ground truth of the Football network and the partition results of SpeakEasy and MDPA. Figure 12 also reveals that the partition results of MDPA are very similar to the ground truth while the partition results of SpeakEasy have many differences from the ground truth. In Figure 12c, Region A contains 15 overlapping nodes that belong to two or three communities, denoted by different colors. A total of 15 nodes are colored yellow and dark blue, which means the yellow community and dark blue communities are exactly the same. Most Region A nodes (13 from 15) belong to the same community in the ground truth (seen as orange nodes in Figure 12a). To further study this community, we found no predominant node, but we found many dominant nodes with the same or similar degrees; therefore, a tendency to retain more than one label within the community surfaced after label propagation in SpeakEasy, which led to the same or basically identical groups of nodes often identified as overlapped communities. While in MDPA, the membership values were propagated instead of the node labels, thereby getting more robustness and avoiding SpeakEasy sameness.

## 5. Conclusions and Discussion

In this research, a novel overlapping community detection algorithm is developed, i.e., the membership degree propagation algorithm (MDPA). The main idea is to propagate the community membership degree according to the difference between global distribution information and local distribution information. After the propagation process, it assigns cluster numbers to a node according to its membership degree. MDPA can substantially reduce both overlapping and non-overlapping community detection problems. In both cases, it does not produce as many partition results as the existing methods did. Moreover, the final partition, as well as the overlapping node recognition result, could be obtained in a single effort based only on the converged membership degree vectors. Hence, it requires a significantly lower computation time and avoids the memory complexities of other programs designed to achieve the same objectives.

To verify the effectiveness of the proposed MDPA, it is applied on synthetic LFR datasets, and 9 real benchmark datasets. Numerical results show that MDPA is competitive compared with other state-of-the-art algorithms. It was one of the top algorithms in terms of both of NMI and S_G_ on LFR datasets. Especially, focused on the overlapping node detection, MDPA is significantly better than other comparison methods on F1 measure. On the real benchmark datasets, compared with other 8 competitive algorithms, MDPA also obtains the best comprehensive performance in terms of Conductance and EQ metrics.

It should be noted that although only the undirected and unweighted networks are discussed in this paper, the proposed MDPA can easily be extended to directed and/or weighted networks just by adding a directed weighting factor into the superscript of *e* in the probability formula (Equation (7)) of the membership degree propagation process. In other words, MDPA has strong adaptability for different kinds of community detection problems, such as social network partition, biomarker detection in bionetworks, and epidemic spreading.

## Figures and Tables

**Figure 1 entropy-23-00015-f001:**
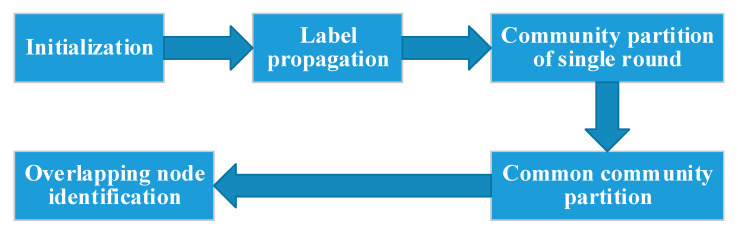
General framework of label propagation methods.

**Figure 2 entropy-23-00015-f002:**
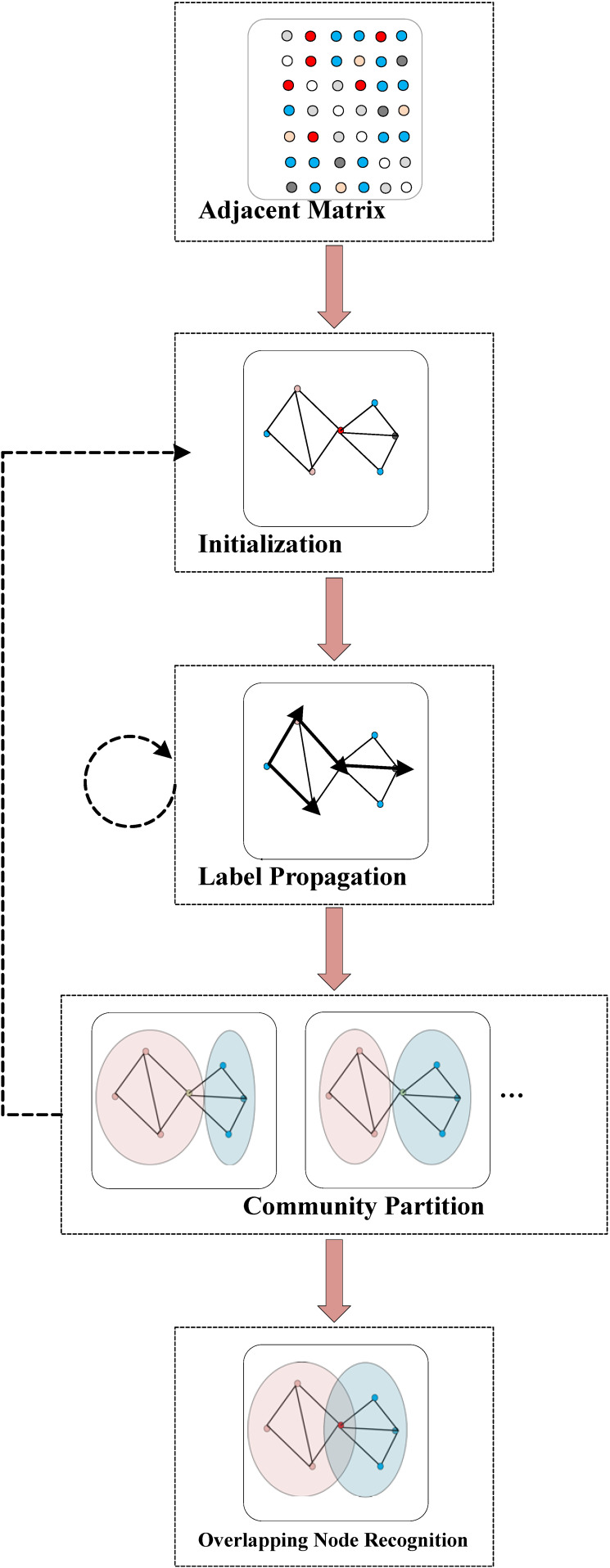
The flowchart of SpeakEasy starts from a network (represented by an adjacent matrix), followed by 4 main steps: (1) initializing for node buffers, (2) label propagating iteratively, and (3) community partitioning when the iteration converges. Steps 1–3 are repeated *n* times. (4) Overlapping nodes and identifying according to obtained *n* partition results.

**Figure 3 entropy-23-00015-f003:**
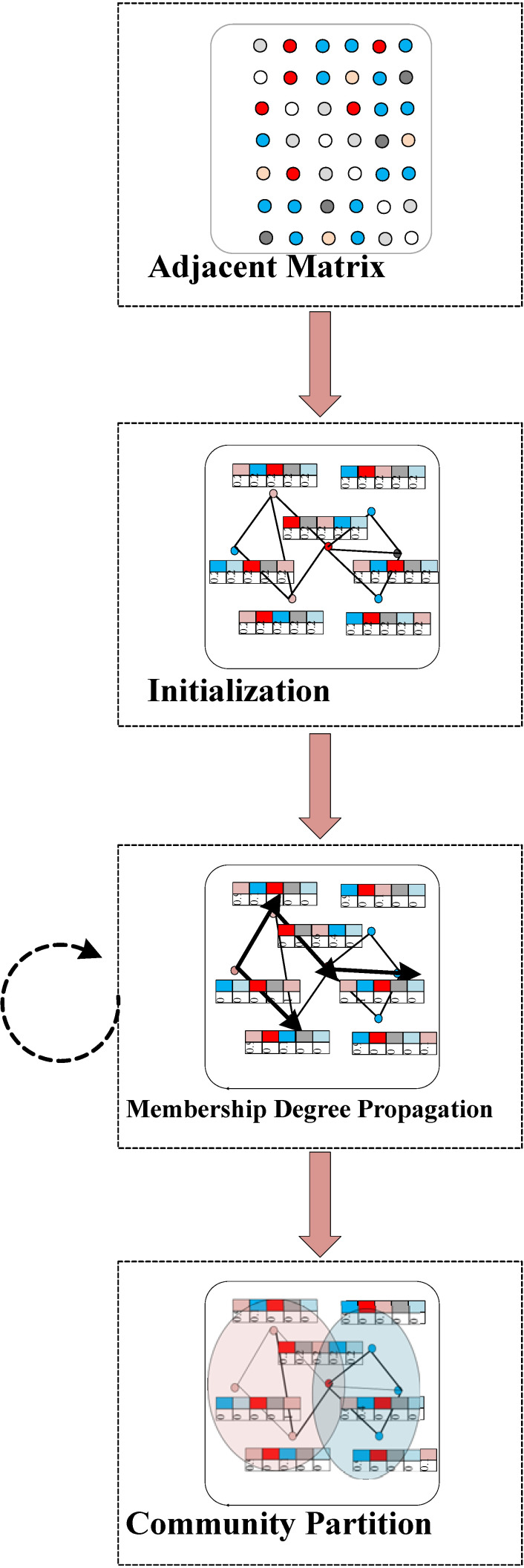
The flowchart of MDPA starts from a network (represented by an adjacent matrix), followed by 3 main steps: (1) initializing for node buffer and membership pairs, (2) membership degree propagating iteratively, and (3) community partitioning when the iteration converges (overlapping nodes are identified simultaneously).

**Figure 4 entropy-23-00015-f004:**
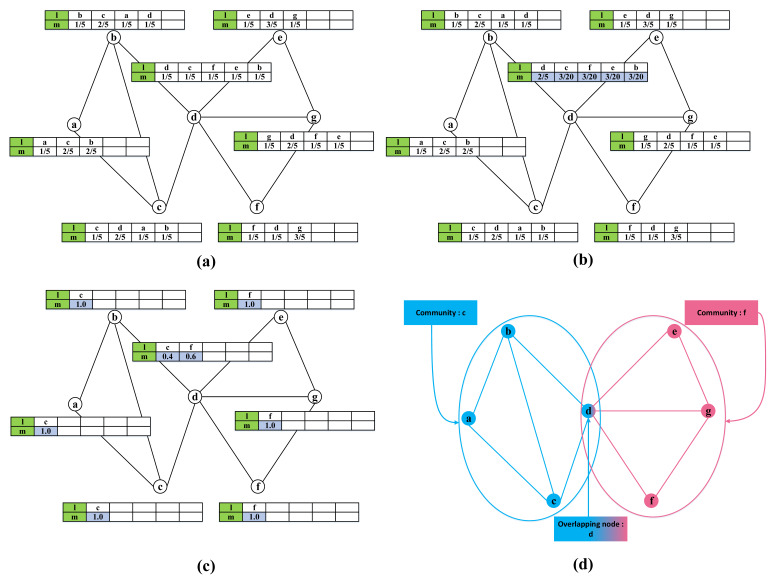
Schematic diagram of MDPA. (**a**) initialization, (**b**) membership degree propagation, (**c**) final state of membership degree propagation, and (**d**) result of community partition. In the green columns of the two-row illustrations, the *l* on the top row represents the cluster label, and the *m* on the bottom row represents the membership degree belonging to a corresponding cluster.

**Figure 5 entropy-23-00015-f005:**
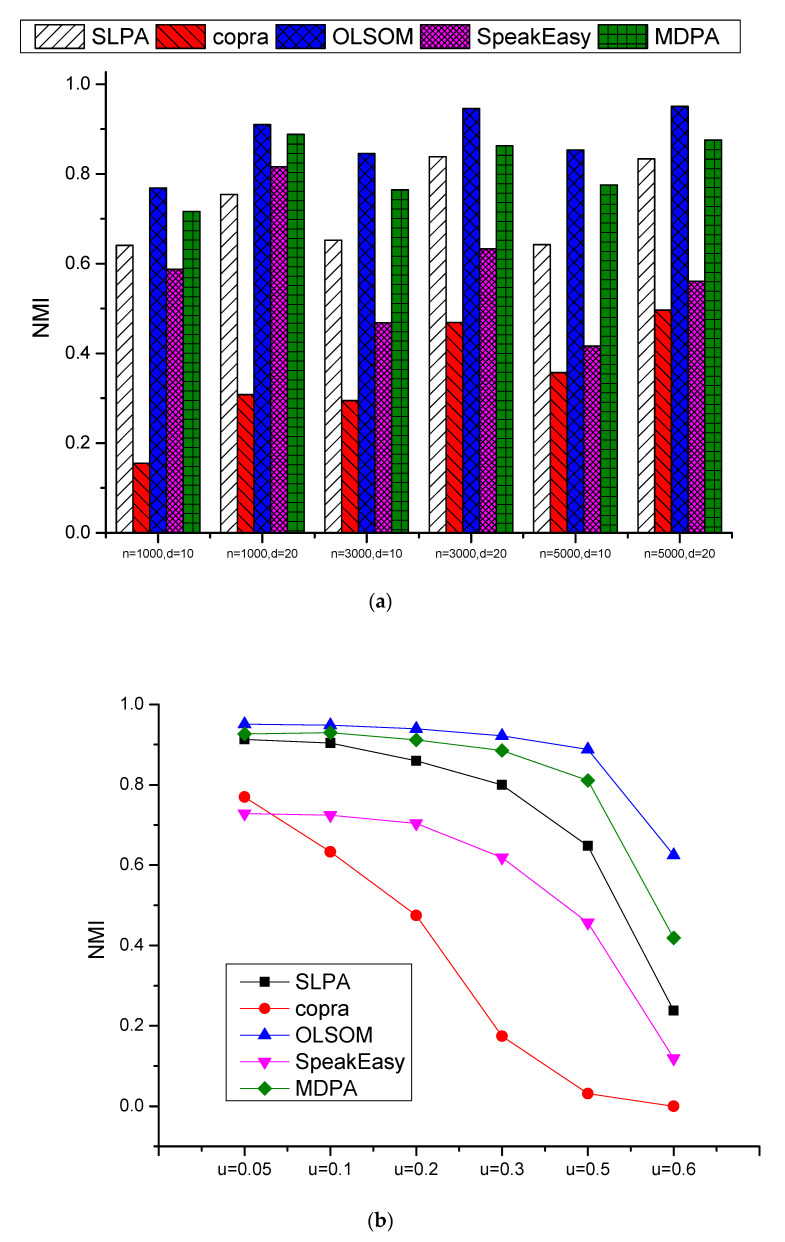
Comparison results of NMI on LFR datasets: (**a**) the average NMI values of different *μ*, *O_n_* and *O_m_* for specified n and d combinations, and (**b**) the average NMI values of different *n*, *d*, *O_n_* and *O_m_* for specified *μ* values.

**Figure 6 entropy-23-00015-f006:**
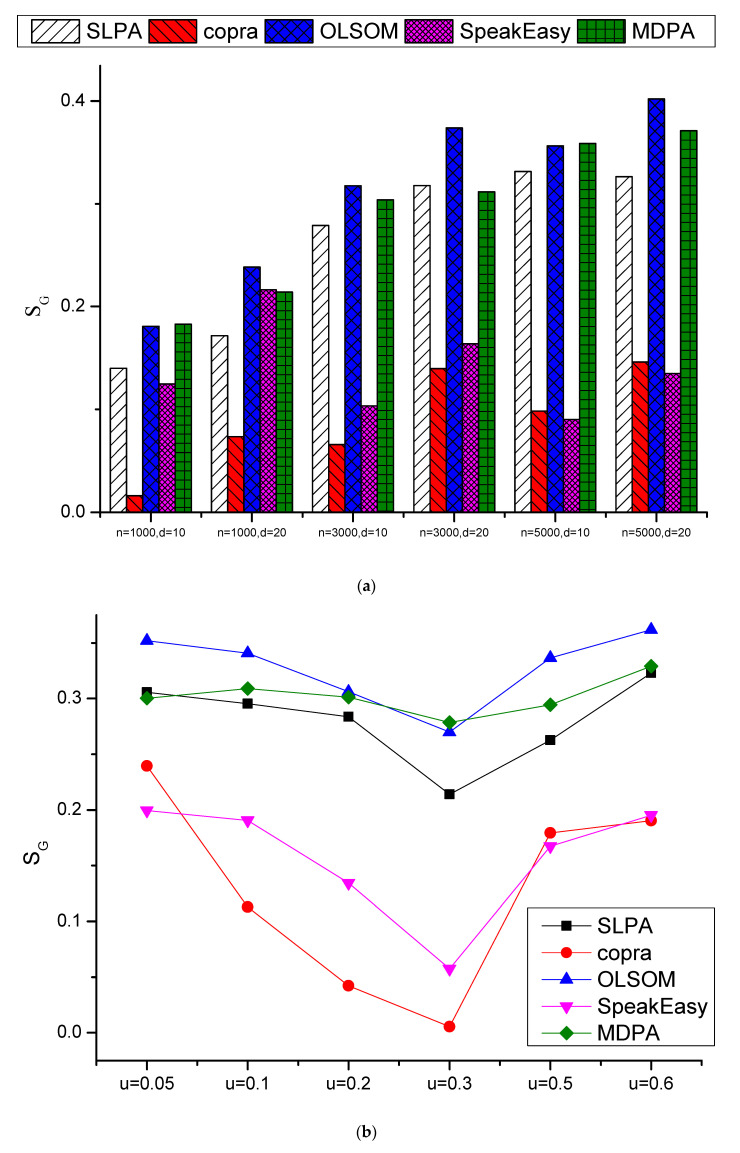
Comparison results of *S_G_* on LFR datasets: (**a**) the average *S_G_* values of different *μ*, *O_n_* and *O_m_* for specified *n* and *d* combinations, and (**b**) the average *S_G_* values of different *n*, *d*, *O_n_* and *O_m_* for specified *μ* values.

**Figure 7 entropy-23-00015-f007:**
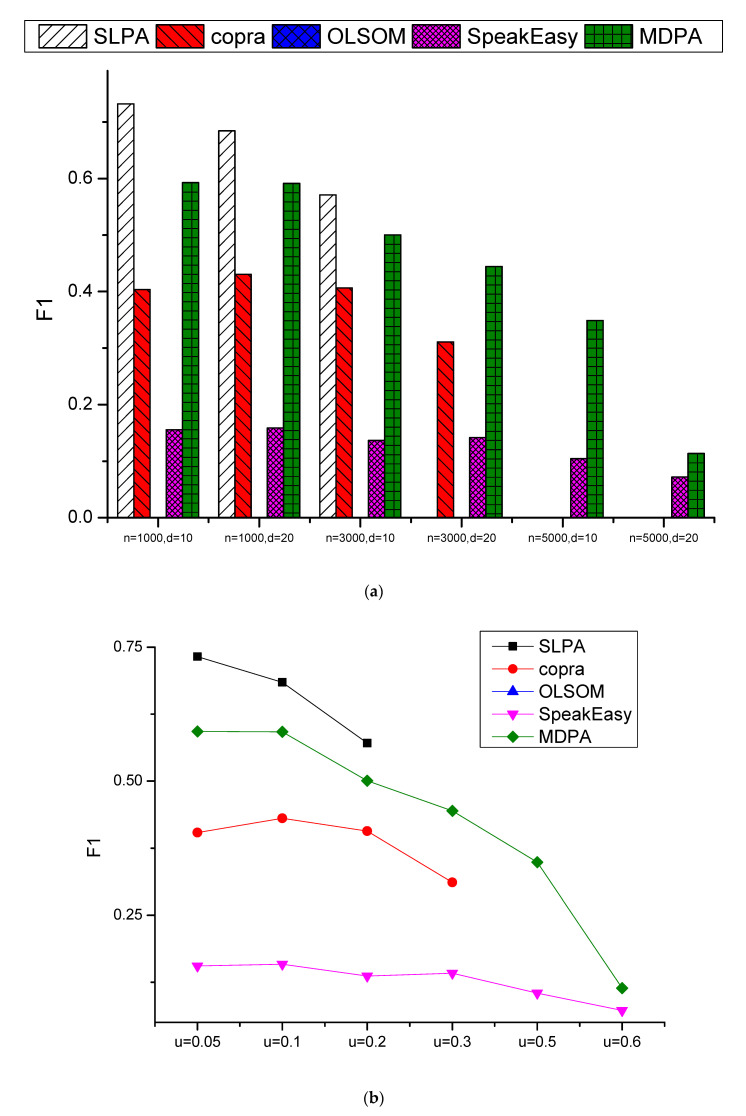
Comparison results of F1 on 144 LFR datasets: (**a**) the average F1 values of different *μ*, *O_n_* and *O_m_* for specified *n* and *d* combinations, and (**b**) the average F1 values of different *n*, *d*, *O_n_* and *O_m_* for specified *μ* values.

**Figure 8 entropy-23-00015-f008:**
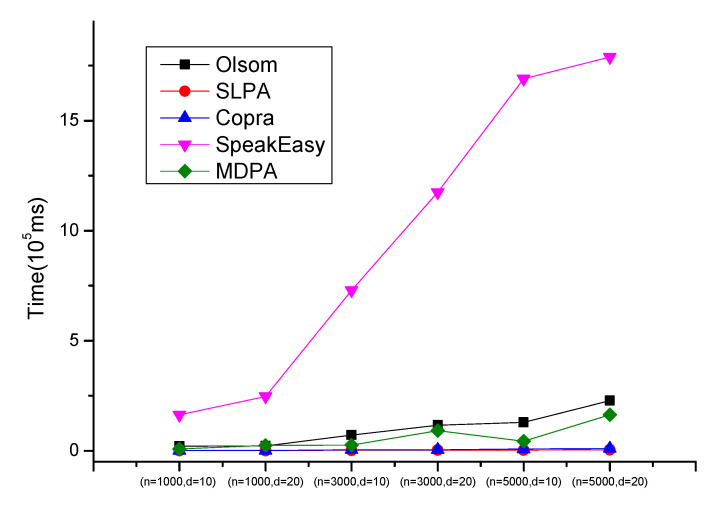
Comparison results of execution time on Lancichinetti–Fortunato–Radicchi (LFR) datasets.

**Figure 9 entropy-23-00015-f009:**
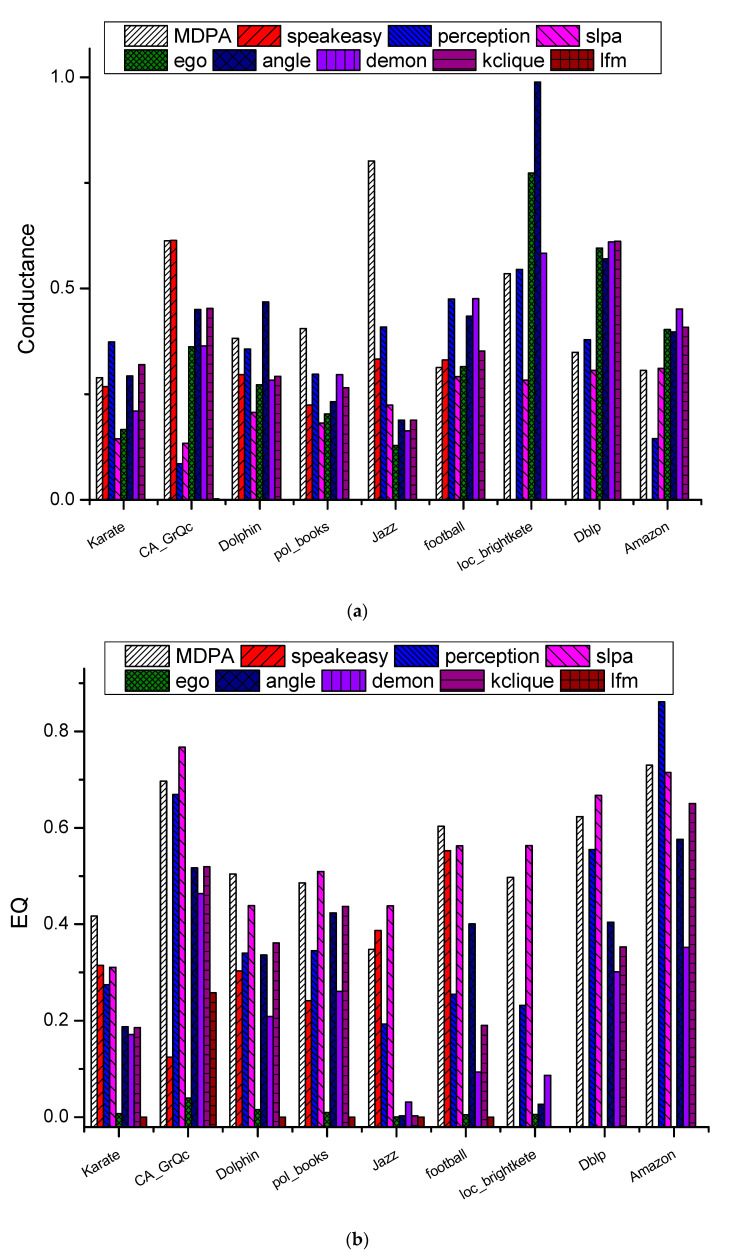
Comparison results of (**a**) Conductance and (**b**) Modularity EQ on nine real datasets.

**Figure 10 entropy-23-00015-f010:**
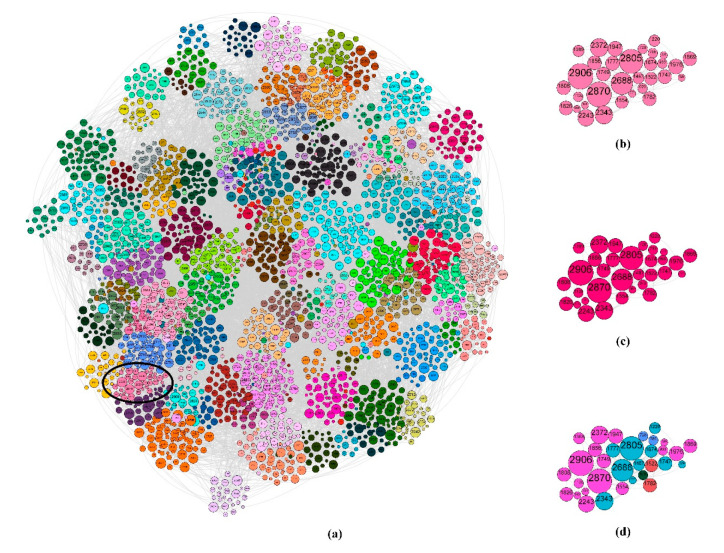
Visualization of LFR network where *n* = 3000, *d* = 10, *μ* = 0.4 and *O_m_* = 6. (**a**) All nodes are colored according to the generated communities, and the nodes in the same community have the same color; (**b**) a community in (**a**); (**c**) the nodes in (**b**) are recolored according to the partition result of MDPA, and (**d**) the nodes in (**b**) are recolored according to the partition results of SpeakEasy.

**Figure 11 entropy-23-00015-f011:**
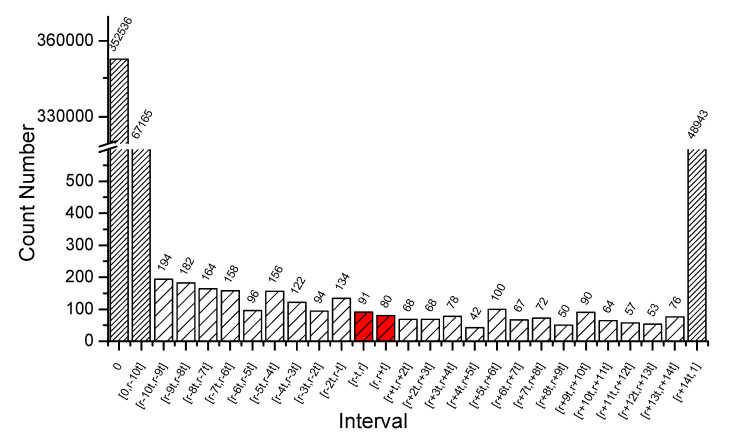
Distribution of the weight defined by Equation (1) in SpeakEasy. *r* = 0.0057 represents the weight threshold in [36,55], and *t* = 0.0001 is the interval step.

**Figure 12 entropy-23-00015-f012:**
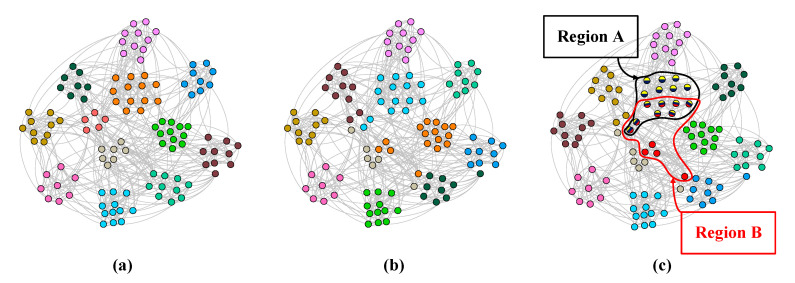
Visualization of Football. (**a**) The Football network has 12 communities and the nodes in the same community have the same color; (**b**) the partition result of MDPA, which has 11 communities and no overlapping nodes, and (**c**) the partition result of SpeakEasy, which has 13 communities and 15 overlapping nodes in Region A.

**Table 1 entropy-23-00015-t001:** Global distribution of all labels.

Label	*a*	*b*	*c*	*d*	*e*	*f*	*g*
Global distribution	3/35	5/35	6/35	10/35	3/35	3/35	5/35

**Table 2 entropy-23-00015-t002:** The local distribution in the neighbor buffers of node *d*.

Label	*a*	*b*	*c*	*d*	*e*	*f*	*g*
Local distribution	2/25	2/25	3/25	9/25	2/25	2/25	5/25

**Table 3 entropy-23-00015-t003:** Original difference, normalized difference, and corresponding probabilities of each label in the neighbor buffers of node *d*.

Label	*a*	*b*	*c*	*d*	*e*	*f*	*g*
Original difference	−1/175	−11/175	−9/175	13/175	−1/175	−1/175	10/175
Normalized difference	50/24	0	10/24	5	50/24	50/24	105/24
Corresponding probability	0.0316	0.0039	0.0060	0.5831	0.0316	0.0316	0.3122

**Table 4 entropy-23-00015-t004:** Parameter setting of generated 144 test networks.

*n*	*d*	*μ*	*O_n_*	*O_m_*
1000	10	{0.05, 0.1, 0.2, 0.3, 0.4, 0.6}	30	{1, 2, 4, 6, 8}
1000	20	{0.05, 0.1, 0.2, 0.3, 0.4, 0.6}	30	{1, 2, 4, 6, 8}
3000	10	{0.05, 0.1, 0.2, 0.3, 0.4, 0.6}	90	{1, 2, 4, 6, 8}
3000	20	{0.05, 0.1, 0.2, 0.3, 0.4, 0.6}	90	{1, 2, 4, 6, 8}
5000	10	{0.05, 0.1, 0.2, 0.3, 0.4, 0.6}	150	{1, 2, 4, 6, 8}
5000	20	{0.05, 0.1, 0.2, 0.3, 0.4, 0.6}	150	{1, 2, 4, 6, 8}

**Table 5 entropy-23-00015-t005:** Detailed information of the real benchmark datasets.

Network	n	M	Description
Karate [52]	34	78	Social network of friendships between 34 members of a karate club at a US university in the 1970s.
Dolphins [53]	62	159	An undirected social network of frequent associations between 62 dolphins in a community living off Doubtful Sound, New Zealand.
Pol. Books [54]	105	441	A network of books about US politics published around the time of the 2004 presidential election and sold by the online bookseller Amazon.com. Edges between books represent frequent co-purchasing of books by the same buyers.
Football [4]	115	613	Network of American football games between Division IA colleges during regular season Fall 2000.
Jazz [55]	198	2742	List of edges of the network of Jazz musicians.
CA-GrQc [56]	5242	14,496	Collaboration network of Arxiv General Relativity category. There is an edge if authors coauthored at least one paper.
Brightkite [57]	58,228	214,078	Brightkite was once a location-based social networking service provider where users shared their locations by checking-in.
DBLP [58]	317,080	1,049,866	The DBLP computer science bibliography provides a comprehensive list of research papers in computer science.
Amazon [58]	334,863	925,872	Network was collected by crawling Amazon website. It is based on the ‘Customers Who Bought This Item Also Bought’ feature of the Amazon website.

The first column lists the dataset name and corresponding reference; the second gives the number of nodes in networks; the third shows the number of edges in networks, and the fourth gives a brief description of the dataset.

**Table 6 entropy-23-00015-t006:** Average EQ and Conductance of different methods on 9 real datasets.

Methods	SpeakEasy	Perception	SLPA	ego	Angle	Demon	Kclique	lfm	MDPA
Ave Conductance	0.344	0.341	0.231	0.358	0.447	0.382	0.361	0.0004	0.444
Ave EQ	0.320	0.414	0.552	0.012	0.319	0.219	0.337	0.0437	0.545

**Table 7 entropy-23-00015-t007:** Confusion matrix for SpeakEasy and MDPA.

SpeakEasy	Predicted Yes	Predicted No	MDPA	Predicted Yes	Predicted No
Actual Yes	88	2	Actual Yes	70	20
Actual No	2390	520	Actual No	221	2689

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
