# Peer review of "Overlapping Community Detection Based on Membership Degree Propagation"

_entropy, 2020, doi:10.3390/e23010015_

Round 1
Reviewer 1 Report
This paper proposes a new algorithm to detect overlapping communities in complex networks. The algorithm is based on label propagation method and intend to partitioning by membership degree propagation. The method can do partition and the overlapping identification in once around.
The authors introduces well the algorithm, but it lack a discusion about some possible parameters: B (max commununity number for a node, they just set B=5) , r (threshold for cut excedents B communities r=1/NI) , they don't discuss about the impact of random initialization of buffer (B neighbors) when they propose to once only execution.
The comparasion with others methods in literature is poor, most methods are avaible for execute any network for improve this comparative study. Mainly for sintetic networks as ground truth and for running time.
The extensive comparision with only Speakeasy is the main weakness of this work for me.
Some minor issues:
Table 3 - some labels are in uppercase and other in lowercase
Algorithm 1 - (line 6 d(v), d(v) <- ...) (?)
Algorithm 1 - in line 3 te correct symbol is \in ?
Lines 361, 366, 373, 388 the word ref appear before the reference number
Figure 5 - Will be easier if the markers was the same for the same algorithm in these two graphics.
Figure 9 - plot using line in this graphic no make sense when x-axis are independent datasets
Other minor issues can be reviewd by the authors in possible revision.
Reviewer 2 Report
Community structure is a ubiquitous property of complex networks. The authors propose MDPA, an overlapping community detection algorithm that overcome the drawback of of speak easy (a label propagation algorithm). The manuscript is well written and methodologically sound. These are strong points of the paper, which makes it a good candidate for a publication in the journal. However, I have a few issues, which needs to be resolved prior to publication:
First of all, I think that the paper can gain in momentum if the importance of overlapping nodes in real-world networks is highlighted. Putting more in context the study might further improve the quality of the manuscript. I suggest to refer to the following literature on the subject in order to extend their discussion:
- Kudelka, E. Ochodkova, S. Zehnalova and J. Plesnik, "Ego-zones: Non-symmetric dependencies reveal network groups with large and dense overlaps", Appl. Netw. Sci., vol. 4, no. 1, pp. 81, Dec. 2019.
Kumar, M., Singh, A. & Cherifi, H. An efficient immunization strategy using overlapping nodes and its neighborhoods. In Companion of the The Web Conference 2018 on The Web Conference 2018 1269–1275 (International World Wide Web Conferences Steering Committee, 2018)
Cherifi, H., Palla, G., Szymanski, B. K. & Lu, X. On community structure in complex networks: challenges and opportunities. Appl. Netw. Sci. 4, 1–35 (2019).
Ghalmane, Z., Cherifi, C., Cherifi, H. & El Hassouni, M. Exploring hubs and overlapping nodes interactions in modular complex networks. IEEE Access 8, 79650–79683 (2020).
Taghavian, F., Salehi, M. & Teimouri, M. A local immunization strategy for networks with overlapping community structure. Phys. A 467, 148–156 (2017)
Yang, J. & Leskovec, J. Structure and overlaps of ground-truth communities in networks. ACM Trans. Intell. Syst. Technol: TIST 5, 1–35 (2014)
Evaluation are performed using synthetic networks as well as real-world networks. This is a good point. However, this raises some questions.
Indeed, although LFR is the standard, it is well-known that is far from generating realistic artificial networks. The authors should at least point out its limitations. Please refer to:
B KamiÅ„ski, P PraÅ‚at, F Théberge Artificial Benchmark for Community Detection (ABCD): Fast Random Graph Model with Community Structure - arXiv preprint arXiv:2002.00843, 2020 - arxiv.org
Finally, Assessing the quality of the community structure using NMI and Modularity is quite basic. In recent works it has been established that more sophisticated methods can be used. The authors should look at and discuss at least briefly the following works:
Dao VL., Bothorel C., Lenca P. (2019) Estimating the Similarity of Community Detection Methods Based on Cluster Size Distribution. In: Aiello L., Cherifi C., Cherifi
H., Lambiotte R., Lió P., Rocha L. (eds) Complex Networks and Their Applications VII. COMPLEX NETWORKS 2018. Studies in Computational Intelligence, vol 812. Springer, Cham
Malek Jebabli, Hocine Cherifi, Chantal Cherifi, Atef Hamouda, Community detection algorithm evaluation with ground-truth data, Physica A: Statistical Mechanics and its Applications, Volume 492, 2018, Pages 651-706, ISSN 0378-4371, https://doi.org/10.1016/j.physa.2017.10.018.
- Ghasemian, H. Hosseinmardi and A. Clauset, "Evaluating Overfit and Underfit in Models of Network Community Structure," in IEEE Transactions on Knowledge and Data Engineering. doi: 10.1109/TKDE.2019.2911585
I hope these remarks will help the authors to improve their manuscript that I enjoyed reading.
Details: I suggest to check the text and change verbs in past and future time to present. Indeed, it is better to use present time as much as possible
Correct the Typos such as:
166 the label is. This process is performed iteratively, until it is converged. ???
167 3. Community partition. After the label propagation process is converged,????
Reviewer 3 Report
Please integrate some reference to other techniques of community detection
and proof read the English
Community elicitation from co-occurrence of activities
Mengoni, P., Li, Y., , DOI: 10.1016/j.future.2019.10.046Author Response
Please see the attached file.

Round 2
Reviewer 1 Report
The authors did not improve the experiments and they did not compare more extensively the results, the revised version only change references, code and a little paragraph in introduction section. The subject is very interesting, but this approach (label propagation community detection) is explored by many authors and need a more extensive comparison with state-of-the-art methods to be considered. In this work the authors compare only with SpeakEasy algorithm (self implementation), and they say that others methods are not available.
In this work the authors use the originals implementation (https://appliednetsci.springeropen.com/articles/10.1007/s41109-020-00289-9)
This library provide many methods (https://cdlib.readthedocs.io/en/latest/index.html)
I suggest the author present a comparative study.
Reviewer 2 Report
I am not satisfied by the author's answer to my following remark:
Finally, Assessing the quality of the community structure using NMI and Modularity is quite basic. In recent works it has been established that more sophisticated methods can be used. The authors should look at and discuss at least briefly the following works:
Dao VL., Bothorel C., Lenca P. (2019) Estimating the Similarity of Community Detection Methods Based on Cluster Size Distribution. In: Aiello L., Cherifi C., Cherifi
H., Lambiotte R., Lió P., Rocha L. (eds) Complex Networks and Their Applications VII. COMPLEX NETWORKS 2018. Studies in Computational Intelligence, vol 812. Springer, Cham
Malek Jebabli, Hocine Cherifi, Chantal Cherifi, Atef Hamouda, Community detection algorithm evaluation with ground-truth data, Physica A: Statistical Mechanics and its Applications, Volume 492, 2018, Pages 651-706, ISSN 0378-4371, https://doi.org/10.1016/j.physa.2017.10.018.
- Ghasemian, H. Hosseinmardi and A. Clauset, "Evaluating Overfit and Underfit in Models of Network Community Structure," in IEEE Transactions on Knowledge and Data Engineering. doi: 10.1109/TKDE.2019.2911585
You cannot say:
Many thanks for the referee’s valuable comments. But according to our experimental setup, our results should be compared with the results reported in other literatures. For this limitation, we could not use those metrics in the referee mentioned literatures.
Indeed, some metrics are quite easy to compute. But what is more important even if you concentrate on basic metric, you must discuss their limitations and make the reader aware that alternative methods of evaluation have been proposed recently to overcome their drawbacks
Round 3
Reviewer 1 Report
The experiments were significantly improved, the authors answered the reviewers' questions and brought more comparative results to the work. Although the algorithm did not achieve the best results in relation to the others algorithms in many scenarios, the methodology is interesting.